# OpenReview forum: "Instruction Following by Principled Attention Boosting of Large Language Models"
_ICLR.cc/2026/Workshop/Sci4DL — Sci4DL 2026_

### Official Review · Reviewer_B9Pb · 2026-02-08

**Fit:** 3
**Significance:** 2
**Confidence:** 2

**Summary:**

The paper reframes instruction following in large language models as a competition between instruction-driven and context-driven influences and formalizes this through there Logicbreaks framework. It studies a simple inference-time intervention (uniformly biasing attention toward instruction tokens) and then theoretically characterizes when this bias improves instruction adherence versus when it degrades contextual reasoning. Based on this, they propose Instaboost, a (light) attention-steering method that achieves strong instruction-following performance across diverse tasks.

**Strengths:**

- Intervention has strong empirical performance and minimal overhead
- Evaluation shows improved instruction adherence and preserves fluency and relevance
- Framing unifies prior attention-steering methods and yields interpretable theoretical guarantees

**Suggestions:**

- Strong theoretical assumptions (e.g., sharp applicability gaps, uniform value scaling)
- Evaluation relies heavily on LLM-as-judge metrics with limited numeric reporting and unclear tuning of the boost strength
- Additional ablations (e.g., layer/head or instruction-span selection) and standardized compliance benchmarks would strengthen the contribution

---

### Official Review · Reviewer_92jh · 2026-02-28

**Fit:** 2
**Significance:** 2
**Confidence:** 2

**Summary:**

This paper studies training-free instruction-following improvements via attention steering. It argues that prior attention-steering methods (PASTA, Spotlight) demonstrate that more attention to instructions can help, but lack a mechanistic account of when this helps versus when it destroys task relevance by suppressing needed context. The authors formalize instruction following using the Logicbreaks abstraction as a competition between instruction-induced rules and context-derived competing rules, with attention controlling the relative influence of these rule sets. In this framework, adding an additive bias to instruction-key attention logits increases instruction-rule influence multiplicatively/exponentially relative to competing rules, making it harder for the context to override instruction-consistent updates. However, it also creates an “over-focus” regime when benign competing rules are suppressed too much. Guided by this, they propose InstABoost, a simple intervention that adds a constant bias B to all pre-softmax attention scores targeting instruction-key positions across all layers and heads (equivalently multiplying those keys’ softmax numerators by $e^B$).
Experiments on Meta-Llama-3-8B-Instruct across 15 tasks report that the proposed approach is competitive with or outperforms compared baselines (including instruction-only prompting, latent steering variants, and attention methods).

**Strengths:**

1. The main strength is the attempt to put attention steering on a principled footing: the rule-competition framing makes the instruction-adherence vs relevance tradeoff explicit and yields qualitative predictions (robustness gains from boosting; suppression/over-focus when competing benign rules are underweighted) that align with the reported empirical failure mode for Spotlight.

2. The proposed approach is minimal and interpretable: a single global knob B applied uniformly (no head-selection profiling as in PASTA, no state-dependent enforcement target as in Spotlight).

**Suggestions:**

1. The evaluation depends heavily on model-based judges (Gemini 2.0 Flash for fluency/relevance; Llama Guard for harmfulness in jailbreak settings). Given that attention steering can change style and refusal patterns in subtle ways, it would strengthen the paper to include either multi-judge agreement/sensitivity checks (different judge models, prompt variants) or a small human-eval slice on the core “relevance degradation” claim to rule out judge artifacts.

2. The framework assumes a cleanly identified instruction span. In practice, instruction boundaries can be ambiguous with chat templates, tool messages, system vs developer vs user content, and multi-instruction prompts. It would be helpful to clarify exactly what tokens are boosted in the main experiments and add an ablation on mis-specified spans (e.g., boosting too much prefix, or missing part of the instruction) to show robustness.

3. Since the core conceptual contribution is the robustness-relevance tradeoff as a function of boost strength, it would help to provide a more task-agnostic guideline for choosing B/M (or a calibration procedure) and to report variability across tasks/models, including cases where InstABoost harms relevance or factuality even when fluency remains high.

---

### Official Review · Reviewer_8Zxd · 2026-03-02

**Fit:** 2
**Significance:** 2
**Confidence:** 1

**Summary:**

This paper formalizes instruction following as a competition between instruction and context rules using the Logicbreaks framework. The authors propose InstABoost, a training-free intervention that applies a constant additive bias to instruction-key attention logits.

**Strengths:**

I don't really follow this line of work so I'm keeping my review brief. The paper proses a very simple additive bias to help instructions "win" without breaking the model, which is nice. The authors conduct a big empirical sweep across two model families which seems very thorough.

**Suggestions:**

I have a few questions that might be useful for the authors when writing their full version

1. How does the performance of InstABoost scale with the length of the instruction prompt, given that $k_t$ (competing rules) grows with the rollout length?

2. Did you observe any specific layers where the additive bias was significantly more effective than others, or is a uniform application across all layers strictly necessary?

---

### Meta-Review · Area_Chair_574X · 2026-02-28

**Recommendation:** Accept

**Metareview:**

This seems suitable to this workshop. I'll note that, for this workshop, I don't really care about the empirical benefits as much as the fact that we come away feeling we've learned something new about deep nets.

---

### Decision · Program_Chairs · 2026-03-02

Accept